# Positive impact of a faecal-based screening programme on colorectal cancer mortality risk

**Gemma Ibáñez-Sanz**[1,2,3,4], **Núria Milà**[4,5,6], **Carmen Vidal**[5,6], **Judith Rocamora**[1,4], **Víctor Moreno**[1,3,4,7], **Rebeca Sanz-Pamplona**[1,3,4], **Montse Garcia**[4,5,6]*, on behalf of the MSIC-SC research group¶

**1** Oncology Data Analytics Program (ODAP), Catalan Institute of Oncology-IDIBELL, L'Hospitalet de Llobregat, Barcelona, Spain, **2** Gastroenterology Department, Bellvitge University Hospital -IDIBELL, L'Hospitalet de Llobregat, Barcelona, Spain, **3** Colorectal Cancer research group, ONCOBELL Programme, Institut d'Investigació Biomèdica de Bellvitge (IDIBELL), L'Hospitalet de Llobregat, Barcelona, Spain, **4** CIBER Epidemiology and Public Health (CIBERESP), Madrid, Spain, **5** Cancer Screening Unit, Prevention and Control Programme, Catalan Institute of Oncology-IDIBELL, L'Hospitalet de Llobregat, Barcelona, Spain, **6** Early Detection of Cancer Research Group, EPIBELL Program, Bellvitge Biomedical Research Institute, L'Hospitalet de Llobregat, Barcelona, Spain, **7** Department of Clinical Sciences, Faculty of Medicine and Health Sciences, University of Barcelona, Barcelona, Spain

¶ Membership of the MSIC-SC research group is listed in the Acknowledgments.
* mgarcia@iconcologia.net

**Data Availability Statement:** The data are publicly available at: https://doi.org/10.34810/data119.

**Funding:** The author(s) received no specific funding for this work.

## Abstract

### Introduction

The effectiveness of colorectal cancer (CRC) screening programs is directly related to participation and the number of interval CRCs. The objective was to analyse specific-mortality in a cohort of individuals invited to a CRC screening program according to type of CRC diagnosis (screen-detected cancers, interval cancers, and cancers among the non-uptake group).

### Material and methods

Retrospective cohort that included invitees aged 50–69 years of a CRC screening program (target population of 85,000 people) in Catalonia (Spain) from 2000–2015 with mortality follow-up until 2020. A screen-detected CRC was a cancer diagnosed after a positive faecal occult blood test (guaiac or immunochemical); an interval cancer was a cancer diagnosed after a negative test result and before the next invitation to the program (≤24 months); a non-uptake cancer was a cancer in subjects who declined screening.

### Results

A total of 624 people were diagnosed with CRC (n = 265 screen-detected, n = 103 interval cancers, n = 256 non-uptake). In the multivariate analysis, we observed a 74% increase in mortality rate in the group with interval CRC compared to screen-detected CRC adjusted for age, sex, location and stage (HR: 1.74%, 95% CI:1.08–2.82, P = 0.02). These differences were found even when we restricted for advanced-cancers participants. In the stratified

**Competing interests:** The authors have declared that no competing interests exist.

**Abbreviations:** CRC, colorectal cancer; gFOBT, guaiac-based faecal occult blood test; FIT, faecal immunochemical test.

analysis for type of faecal occult blood test, a lower mortality rate was only observed among FIT screen-detected CRCs.

## Conclusion

CRC screening with the FIT was associated with a significant reduction in CRC mortality.

## Introduction

Faecal occult blood test (FOBT) screening followed by a colonoscopy has been demonstrated to decrease colorectal cancer (CRC) mortality [1] and has been implemented in most developed countries as the preferable method for CRC screening [2, 3]. Population-screening programs aim to reduce CRC mortality thorough early detection [4]. For this reason, it is important to detect the largest number of existing tumours in the population before their clinical occurrence. Additionally, FOBTs can detect some preneoplastic polyps that can be removed during colonoscopy, thereby reducing the incidence of cancer [2]. To achieve this goal, the screening program needs to be well organized and requires high uptake. Participation rates in population-based organized CRC screening programmes vary widely across countries [5]. A higher participation rate is observed in FOBT screening than colonoscopy [6], especially among subjects invited to faecal immunochemical test (FIT) screening compared to those invited to guaiac faecal occult blood test (gFOBT) screening [7].

Screening attempts to detect early-stage cancers to achieve a better prognosis and improved mortality. Consequently, the monitoring of interval cancers is a key performance indicator of an organized population-based screening programme and provides a mechanism to evaluate the impact of the program on CRC mortality in the target population. In a CRC screening programme, there are three distinct categories of cancer diagnoses. The non-uptake cancers (those among individuals who decline the screening test and are later diagnosed with a CRC), interval cancers (a cancer detected after a negative FOBT and before the next invitation is due [8]) and screen-detected cancers. Interval cancers may either be due to a false-negative FOBT result or fast-growing (high microsatellite instability and CpG island methylator phenotype (CIMP)-high) [9] tumours that have developed between screening rounds. Consequently, the effectiveness of the program also depends on the ability of the screening test to detect cancers. The gFOBT has been replaced by the FIT because of its higher sensitivity and specificity [2, 5, 10, 11]. However, the FIT, as a screening test, has a sensitivity of 86% to detect CRC [1] so interval cancers are inevitable but their number should be as small as possible to ensure screening effectiveness.

The aim of the study was to quantify the impact on mortality of identifying CRCs through screening compared to interval cancer. We also aimed to compare differences in CRC mortality in clinical-detected cases (interval cancers versus non-uptake cancers).

## Materials and methods

Briefly, Catalan Institute of Oncology (Barcelona) manages a free, public, biennial screening programme using FOBT. From 2000–2010, the gFOBT was used as the screening test (Hema-screenTM). In 2010, the FIT with a cut-off of $\geq$20μg Hb/g faeces (100 ng/mL) (OC-Sensor, Eiken Chemical Co., Japan) replaced the gFOBT as the screening test of choice.

The CRC screening program was launched as a pilot program and was considered an established program by the third round. The median time between invitations was 33 months in the

firsts rounds with gFOBT due to organizational and management constraints. However, in subsequent rounds with the FIT the median time between screening invitations was 24 months.

The target population of the current study (n = 85,000 approximately) includes all men and women aged 50–69 years from l'Hospitalet de Llobregat, Vilafranca del Penedès and Penedès rural (Barcelona, Catalonia, Spain). Exclusion criteria of the screening programme are as follows: gastrointestinal symptoms, familial or personal history of CRC, hereditary CRC syndromes, prior adenomas, or inflammatory bowel disease, colonoscopy in the previous five years or an FOBT within the last two years, terminal disease and severe disabling conditions. Individuals with an invalid mailing address were also excluded. A detailed description of the programme and its quality indicators has been previously described [12, 13].

The study period was from May 2000 to December 2015 with a minimum vital status follow-up of 36 months. The following variables were taken into account in the analysis: sex, age, area-level deprivation, type of FOBT, type of participation (first invitation or successive), number of screening participations, date of CRC diagnosis, details of CRC (location, morphology, TNM staging (8th edition) classified as early (TNM I/II) or advanced (TNM III/IV) stage), and vital status that was identified through the Population Death Register until 1 August 2020. The deprivation level was calculated for primary healthcare areas of the Catalan territory [14]. It is a score index that uses aggregated indicators of income, employment, health, disability and education to generate a scale from 0 (least deprived) to 100 (most deprived). The scores were recoded into terciles to enable comparisons between individuals living in the most and least deprived area.

The CRC screening programme information system allowed recording: FIT result, number of individual participations, screening test (guaiac or immunochemical), final screening test results (positive, negative, or indeterminate), and colonoscopy findings. Then, this information was linked with the minimum basic set of hospital discharge data (CMBD-AH), which contains information on procedures and diagnoses for each hospital admission of public hospitals of Catalonia and has reasonable accuracy to identify cancer cases [15]. The two databases were linked to allow classification of subjects into their respective groups of screened-detected, interval, and non-uptake. Participation was defined as having a conclusive FOBT result. All the participants were followed until December 2015 to identify whether they were diagnosed after a FOBT with a CRC (International Classification of Diseases 10th Revision [ICD-10]: C18, C19 and C20: colon or rectum). We did an exhaustive chart review where we went through colonoscopy and surgical reports as well as admission notes to confirm the classification of these individuals, and to recollect study covariates and cause of death.

The CRCs were identified and classified into: 1) Screen-detected cancer: a cancer detected after a positive FOBT (guaiac and immunochemical); 2) Interval cancer: a cancer diagnosed after a negative FOBT result and before the next scheduled screening invitation(8) ($\leq$24 months); 3) Non-uptake cancers: a cancer in subjects who were invited to the screening, but did not participate in it. Although the next scheduled screening was defined to be 24 months after the previous screen, the interval between screening invitations was higher in the earlier rounds with gFOBT. We have performed a sensitivity analysis in gFOBT participants defining interval cancers as those diagnosed $\leq$30 months after the last gFOBT and no differences were observed (see S1 Table and S1 Fig in S1 File)

There were an additional 33 diagnoses of CRC after a FIT positive result (post-colonoscopy CRC); 1 case diagnosed before the first participant's invitation, 2 diagnoses in subjects with inherited CRC syndromes, and 208 were diagnosed more than 24 months after screening FOBT analysis (individuals with a negative FOBT result who did not participate in successive screening); these diagnoses were excluded from the analysis.

The location of the CRC was defined according to its anatomic distribution: proximal, distal, and rectum. CRCs diagnosed in the cecum, ascending colon, hepatic flexure and transverse colon were considered proximal or right-sided. Distal or left-sided cancers were those diagnosed in the splenic flexure, the descending colon, sigmoid colon and the recto-sigmoid junction. Anal and appendix cancers were excluded.

Our CRC screening programme follows the Public Health laws and the Organic Law on Data Protection. All procedures performed in the study involving data from human participants were in accordance with the ethical standards of the institutional research committee, and with the 1964 Helsinki Declaration and its later amendments or comparable ethical standards. No informed consent was requested to the participating individuals, since this study was based on anonymized data routinely collected. The study protocol was approved by the Ethics Committee for Clinical Research (PR234/16).

Exact test of Fisher or $\chi^2$-test were used for categorical variables and 2 categories $t$-test or the non-parametric Mann Whitney $U$ test were used for quantitative variables. Overall mortality after CRC diagnosis was compared by Kaplan-Meier estimation, long-rank test and Cox proportional hazard ratios to date of death or censoring resulting from the end of the study period (1 August 2020). In the survival analysis, non-CRC deaths or unknown cause of death (n = 6) were censored. We performed a sensitivity analysis in participant subjects (screening and interval groups) to correct for lead-time bias. In this analysis the survival time was measured from the date of the last FOBT performed to date of death or censoring. Statistical analysis was carried out using R statistical software (R Foundation for Statistical Computing, Vienna, Austria).

## Results

Within the study period of 1 May 2000 to 31 December 2015, 539.475 individuals were invited, with an uptake rate that increased from 17.2% in the first round to 45.8% in the last one. Fig 1 presents a flowchart of the population examined, from when they were invited to perform the FOBT to when they were diagnosed with CRC.

A total of 624 people were diagnosed with CRC within the age range (50–69 years) eligible for screening plus 24 months. Of these, 265 (42.5%) were screen-detected, 103 (16.5%) were interval cancers, 256 (41.0%) were diagnosed in individuals who were invited but did not participate (non-uptake group). Interval cancers were detected in ≤12 months of the FOBT performance in 33.0% of the cases (n = 34) and 67.0% (n = 69) of them were diagnosed after 12 months.

### Comparison of all the subgroups of colorectal cancers

Table 1 shows the patients' characteristics as well as the characteristics of the tumour in terms of the location, histologic type, and stage. The 5-year mortality in CRC was 11.8% (95% CI; 7.8–15.7) in screen-detected group, 25.1% (95% CI; 16.1–33.2) in interval group and 34.8% (95% CI; 28.8–40.8) in non-uptake group. With a mean follow-up of 81.2 (SD = 51.6) months across the population, the number of patients who died of CRC was 166 (79.4%) and 37 (17.7%) of other causes.

When comparing the three CRC groups, we observed a 67% (95% CI; 1.04–2.70) increase in mortality rate in the CRC interval group and a two-fold risk (95% CI; 1.51–3.30) of mortality in the non-uptake group compared to the screen-detected CRC adjusted for sex, age, socioeconomic index, location, and TNM stage (Fig 2). Male sex and advanced stage were independent predictors of a higher mortality rate (Fig 2).

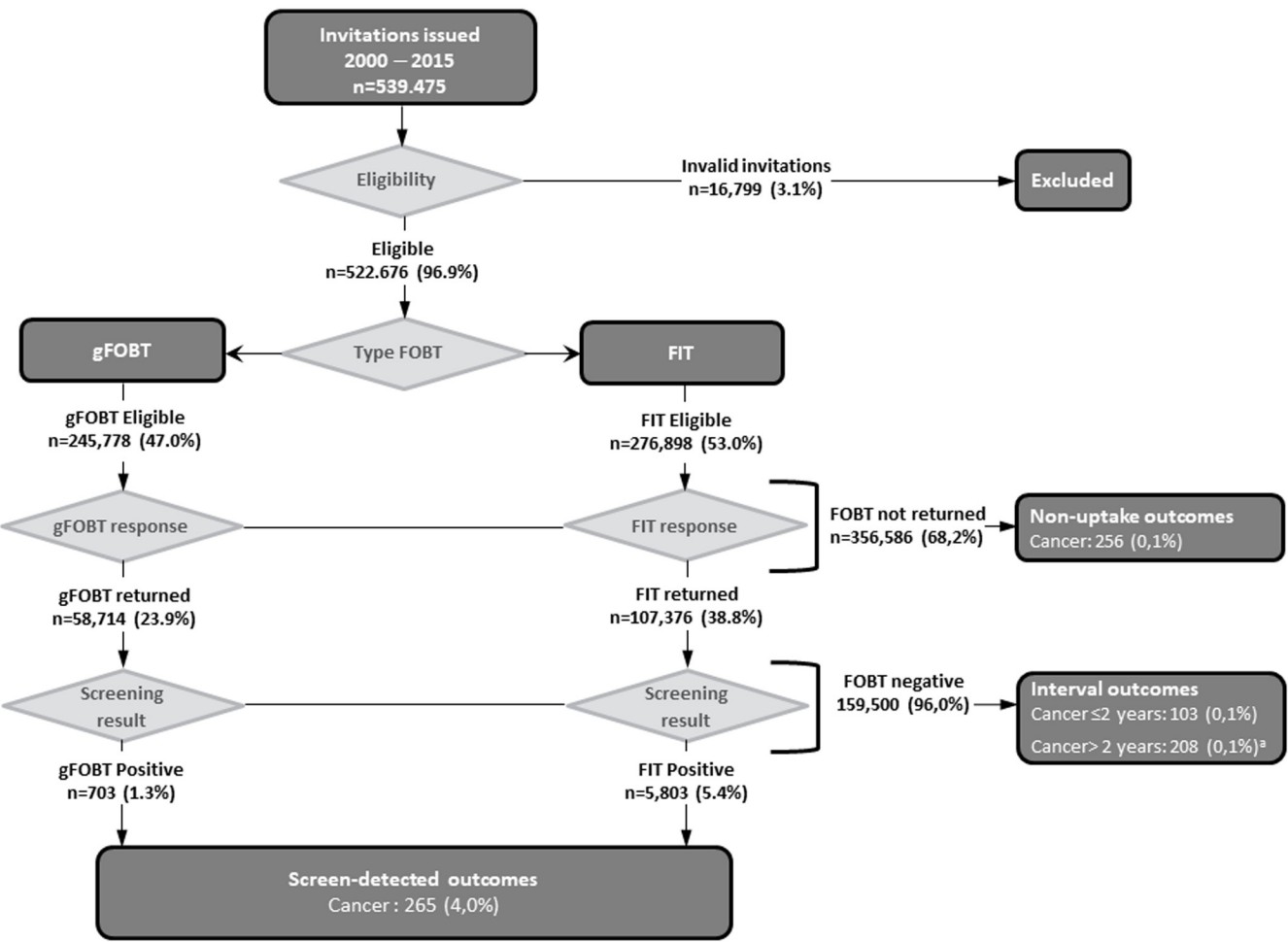

**Fig 1. Colorectal cancers for the 2000–2015 invitee study group.** [a]Colorectal cancer diagnoses >24 months after FOBT analysis were excluded from the analysis. FOBT: faecal occult blood test; gFOBT: guaiac faecal occult blood test; FIT: faecal immunochemical test.

## Participants (screen-detected or interval) vs non-uptake group

Among the non-uptake individuals there were more males than among participants. In fact, there were 128 women in the participants' group (34.8%) and 68 in the non-uptake group (26.6%) (p = 0.04). When compared with the non-uptake group, the participants group were more staged I (35.4% vs 8.44%) and less staged IV (14.1% vs 29.1%) with no differences regarding area-level deprivation, location, and histologic type (Table 1). When these two groups were compared against each other, the non-uptake group had an 84% higher risk of mortality (HR: 1.84%, 95% CI: 1.32–2.55, P = 0.0003) (Table 2).

## Screen-detected vs interval group

When compared with the screen-detected group, interval CRCs were associated with successive screening, gFOBT as the last type of test performed before the diagnoses, rectum and proximal location, advanced TNM stage, and death for CRC, with no differences in sex, age, area-level deprivation, and histologic type (Table 1). It is important to highlight that 41.1% of the screen-detected CRCs were stage I. Among the participants, higher mortality rates were

**Table 1. Characteristics of colorectal cancer according to type of detection.**

| | Screen-detected | Interval | Non-uptake | Screen vs Interval | Interval vs Non-uptake | Participants[a] vs Non-uptake |
|---|---|---|---|---|---|---|
| | *n = 265* | *n = 103* | *n = 256* | P-value | P-value | P-value |
| Sex | | | | 1.00 | 0.14 | 0.037 |
| Female | 92 (34.7%) | 36 (35.0%) | 68 (26.6%) | | | |
| Male | 173 (65.3%) | 67 (65.0%) | 188 (73.4%) | | | |
| Age at diagnosis (mean (SE), years) | 61.9 (5.5) | 62.7 (5.0) | 63.3 (5.6) | 0.17 | 0.33 | 0.008 |
| Age at diagnosis (years) | | | | 0.09 | 0.64 | 0.336 |
| 50–59 | 90 (34.0%) | 25 (24.3%) | 70 (27.3%) | | | |
| 60–72 | 175 (66.0%) | 78 (75.7%) | 186 (72.7%) | | | |
| Socioeconomic Score | | | | 0.25 | 0.01 | 0.027 |
| 0–39 (least deprived) | 10 (3.77%) | 1 (0.97%) | 11 (4.30%) | | | |
| 39–51 | 80 (30.2%) | 27 (26.2%) | 98 (38.3%) | | | |
| 52–100 (most deprived) | 175 (66.0%) | 75 (72.8%) | 147 (57.4%) | | | |
| Last participation | | | | <0.001 | NA | NA |
| Initial | 191 (72.1%) | 43 (41.7%) | NA | | | |
| Successive | 74 (27.9%) | 60 (58.3%) | NA | | | |
| Type of last FOBT | | | | <0.001 | NA | NA |
| gFOBT | 190 (71.7%) | 41 (39.8%) | NA | | | |
| FIT | 75 (28.3%) | 62 (60.2%) | NA | | | |
| Tumour Location | | | | 0.001 | 0.19 | 0.340 |
| Distal | 140 (52.8%) | 34 (33.0%) | 110 (43.0%) | | | |
| Proximal | 61 (23.0%) | 27 (26.2%) | 57 (22.3%) | | | |
| Rectum | 59 (22.3%) | 40 (38.8%) | 82 (32.0%) | | | |
| Missing | 5 (1.9%) | 2 (1.9%) | 7 (2.7%) | | | |
| Histologic type | | | | 0.10 | 0.89 | 0.213 |
| Adenocarcinoma | 180 (67.9%) | 62 (60.2%) | 192 (75.0%) | | | |
| Mucinous | 7 (2.6%) | 5 (4.9%) | 17 (6.6%) | | | |
| Others | 2 (0.8%) | 3 (2.9%) | 7 (2.7%) | | | |
| Missing | 76 (28.7%) | 33 (32.0%) | 40 (15.6%) | | | |
| TNM Stage | | | | <0.001 | 0.04 | <0.001 |
| I | 109 (41.1%) | 19 (18.4%) | 20 (7.8%) | | | |
| II | 54 (20.4%) | 23 (22.3%) | 62 (24.2%) | | | |
| III | 72 (27.2%) | 34 (33.0%) | 86 (33.6%) | | | |
| IV | 28 (10.6%) | 23 (22.3%) | 69 (27.0%) | | | |
| Missing | 2 (0.8%) | 4 (3.9%) | 19 (7.4%) | | | |
| TNM Stage | | | | 0.001 | 0.22 | <0.001 |
| I-II | 163 (61.5%) | 42 (40.8%) | 82 (32.0%) | | | |
| III-IV | 100 (37.7%) | 57 (55.3%) | 155 (60.5%) | | | |
| Missing | 2 (0.8%) | 4 (3.9%) | 19 (7.4%) | | | |
| Death for CRC | | | | 0.001 | 0.41 | <0.001 |
| No | 224 (84.5%) | 71 (68.9%) | 163 (63.7%) | | | |
| Yes | 41 (15.5%) | 32 (31.1%) | 93 (36.3%) | | | |

[a]Participants includes individuals with screen-detected cancers and interval cancers.

SE: standard error; FOBT: faecal occult blood test; gFOBT: guaiac faecal occult blood test; FIT: faecal immunochemical test; CRC: colorectal cancer; NA: not applicable.

observed among patients with interval cancers (HR: 1.74%, 95% CI:1.08–2.82, P = 0.02) (Table 2) than with screen-detected cancers. Interaction analysis between screen-detected

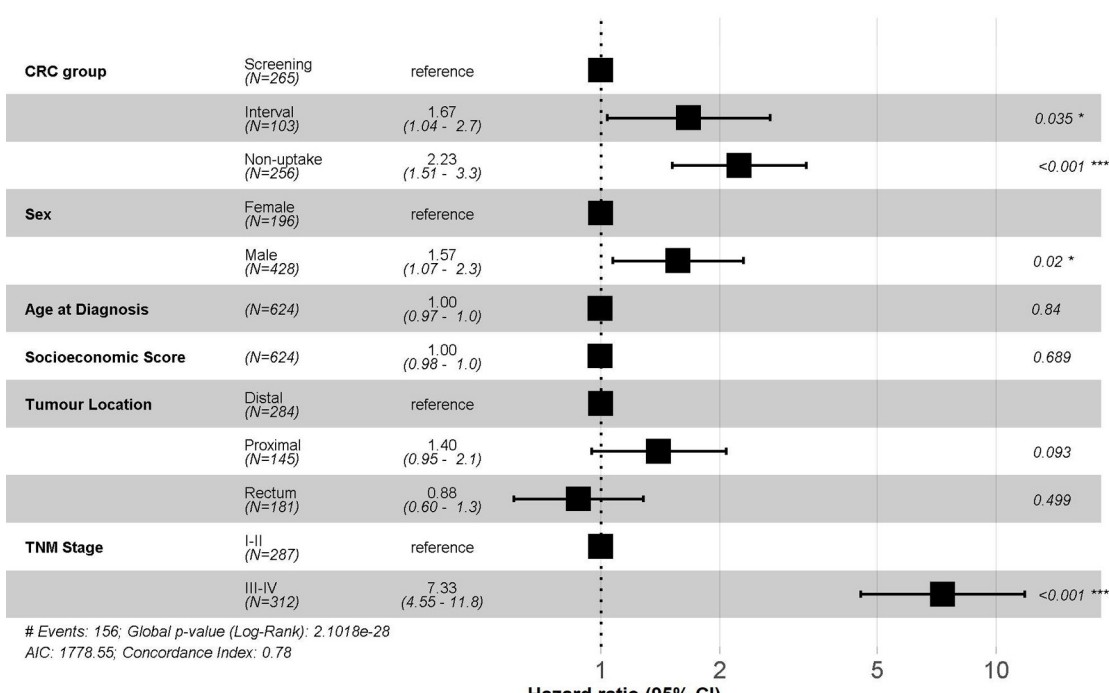

**Fig 2. Multivariate analyses for colorectal cancer mortality comparing all the invitee study group.** [a]Hazard ratios adjusted by the variables shown in this figure. CRC: colorectal cancer.

**Table 2. Multivariate analyses for colorectal cancer mortality according to type of detection.**

| | | SC vs IC vs NC | | SC vs IC | | IC vs NC | | PC vs NC | |
|---|---|---|---|---|---|---|---|---|---|
| | | HR | 95% CI | HR | 95% CI | HR | 95% CI | HR | 95% CI |
| Sex | | | | | | | | | |
| | Female | 1.00 | | 1.00 | | 1.00 | | 1.00 | |
| | Male | **1.57** | **1.07–2.30** | **1.79** | **1.05–3.05** | 1.42 | 0.91–2.20 | **1.57** | **1.07–2.30** |
| Age at diagnosis | | 1.00 | 0.97–1.00 | 1.00 | 0.95–1.04 | 1.00 | 0.97–1.00 | 1.00 | 0.97–1.03 |
| Socioeconomic index | | 1.00 | 0.98–1.00 | 1.00 | 0.98–1.03 | 1.99 | 0.98–1.00 | 1.00 | 0.98–1.01 |
| Tumour location | | | | | | | | | |
| | Distal | 1.00 | | 1.00 | | 1.00 | | 1.00 | |
| | Proximal | 1.40 | 0.95–2.10 | 1.19 | 0.68–2.09 | 1.43 | 0.88–2.30 | 1.43 | 0.97–2.12 |
| | Rectum | 0.88 | 0.60–1.30 | **0.54** | **0.29–0.98** | 1.15 | 0.75–1.80 | 0.91 | 0.62–1.32 |
| CRC group | | | | | | | | | |
| | Screening | 1.00 | | 1.00 | | NA | | 1.00 | |
| | Interval | **1.67** | **1.04–2.70** | **1.74** | **1.04–2.91** | 1.00 | | | |
| | Non-uptake | **2.23** | **1.51–3.30** | NA | | 1.32 | 0.89–2.00 | **1.84** | **1.32–2.55** |
| TNM stage | | | | | | | | | |
| | I-II | 1.00 | | 1.00 | | 1.00 | | 1.00 | |
| | III-IV | **7.33** | **4.55–11.80** | **8.95** | **4.65–17.23** | **5.95** | **3.33–10.60** | **7.68** | **4.77–12.36** |

SC: screen-detected colorectal cancers; IC: interval cancers: NC: cancers in the non-uptake group; PC: cancers in the group of participants (includes individuals with screen-detected and interval cancers; HR: hazard ratio; CRC: colorectal cancer; NA: not applicable.

Associations with $p<0.05$ are shown in bold.

colorectal cancers and interval cancers according with the TNM stage is shown in Table 3. A higher mortality rate in the group of interval cancer with early stage compared to the screening group with the same stage was observed. The S2 Fig in S1 File displays the Kaplan–Meier curves of screen-detected cancers and interval cancers stratified for early (stage I-II) and advanced (stage III-IV). After considering for the lead-time bias, the hazard ratio for interval CRCs was not significant (HR: 1.55, 95% CI:0.96–2.50, P = 0.08).

### Interval vs non-uptake group

When compared with the non-uptake group, the interval cancer group were more likely to live in a more socioeconomically deprived area and to be diagnosed with a staged I cancer. However, no differences in mortality were found (Table 2).

### Stratified analysis according to screening test (gFOBT vs FIT)

The characteristics of the participants in the screening programme according to the type of last FOBT performed are shown in Table 4. Successive screening with any FOBT was associated with interval cancer. We observed a differential distribution in the location of the tumour. Distal location was the most frequent in screened-cancers specially with gFOBT. There was a significant difference in mortality among interval cancers when using the FIT (Fig 3) but not for gFOBT participants (S1A and S1B Fig in S1 File). CRC mortality among FIT participants was 11.2% (95% CI; 6.6–15.6) in screen-detected group and 27.2% (95% CI; 12.0–39.7) in interval group. These differences were found even when restricting for advanced-cancers (Table 3) and after considering for lead-time bias (HR: 2.30, 95% CI: 1.06–5.01, P = 0.04).

## Discussion

In this population-based CRC screening programme study, of the 624 cancers diagnosed in the group offered screening, 265 (42.5%) were detected by FOBT screening, 103 (16.5%) presented after a negative FOBT and before the next invitation, and 256 (41.0%) presented in the non-uptake group.

**Table 3.  Interaction analysis between screen-detected colorectal cancers and interval cancer and TNM stage.**

| Participants in FOBT screening | | |
| --- | --- | --- |
| | **Adjusted OR[a]** | **95% CI** |
| Interval CRC–Stage III-IV | 1.00 | |
| Screened-detected CRC–Stage III-IV | 0.61 | 0.37–1.02 |
| Interval CRC–Stage I-II | 0.16 | 0.05–0.44 |
| Screened-detected CRC–Stage I-II | 0.06 | 0.03–0.13 |
| **Participants in FIT screening** | | |
| | **Adjusted OR[a]** | **95% CI** |
| Interval CRC–Stage III-IV | 1.00 | |
| Screened-detected CRC–Stage III-IV | 0.34 | 0.15–0.77 |
| Interval CRC–Stage I-II | 0.05 | 0.01–0.42 |
| Screened-detected CRC–Stage I-II | 0.03 | 0.01–0.09 |

[a]Adjusted by sex, age, socioeconomic score, and location.

FOBT: faecal occult blood test (guaiac or immunochemical); FIT: faecal immunochemical test; CRC: colorectal cancer.

**Table 4. Characteristics of colorectal cancer among participants according to type of faecal occult blood test.**

| | g-FOBT | | | FIT | | |
|---|---|---|---|---|---|---|
| | Screening | Interval | P-value | Screening | Interval | P-value |
| | *n = 75* | *n = 62* | | *n = 190* | *n = 41* | |
| Sex | | | 0.96 | | | 1.000 |
| Female | 24 (32.0%) | 21 (33.9%) | | 68 (35.8%) | 15 (36.6%) | |
| Male | 51 (68.0%) | 41 (66.1%) | | 122 (64.2%) | 26 (63.4%) | |
| Age at diagnosis (median (SE), years) | 61.7 (5.03) | 62.7 (4.99) | 0.26 | 62.0 (5.65) | 62.8 (5.10) | 0.38 |
| Socioeconomic Score | | | 0.62 | | | 0.69 |
| 0–39 (least deprived) | 0 (0.0%) | 0 (0.0%) | | 10 (5.26%) | 1 (2.44%) | |
| 39–51 | 17 (22.7%) | 11 (17.7%) | | 63 (33.2%) | 16 (39.0%) | |
| 52–100 (most deprived) | 58 (77.3%) | 51 (82.3%) | 0.62 | 117 (61.6%) | 24 (58.5%) | |
| Last participation | | | 0.004 | | | <0.001 |
| Initial | 50 (66.7%) | 25 (40.3%) | | 141 (74.2%) | 18 (43.9%) | |
| Successive | 25 (33.3%) | 37 (59.7%) | | 49 (25.8%) | 23 (56.1%) | |
| Number of FOBT participations | | | 0.013 | | | 0.001 |
| 1 | 50 (66.7%) | 25 (40.3%) | | 141 (74.2%) | 18 (43.9%) | |
| 2 | 12 (16.0%) | 22 (35.5%) | | 37 (19.5%) | 17 (41.5%) | |
| 3 | 10 (13.3%) | 12 (19.4%) | | 12 (6.3%) | 6 (14.6%) | |
| 4 | 3 (4.0%) | 3 (4.8%) | | NA | NA | |
| Tumour Location | | | 0.008 | | | 0.04 |
| Distal | 45 (60.0%) | 22 (35.5%) | | 95 (50.0%) | 12 (29.3%) | |
| Proximal | 15 (20.0%) | 12 (19.4%) | | 46 (24.2%) | 15 (36.6%) | |
| Rectum | 15 (20.0%) | 26 (41.9%) | | 44 (23.2%) | 14 (34.1%) | |
| Missing | 0 (0.0%) | 2 (3.2%) | | 5 (2.6%) | 0 (0.0%) | |
| TNM Stage | | | 0.03 | | | 0.05 |
| I | 28 (37.3%) | 9 (14.5%) | | 81 (42.6%) | 10 (24.4%) | |
| II | 15 (20.0%) | 13 (21.0%) | | 39 (20.5%) | 10 (24.4%) | |
| III | 22 (29.3%) | 23 (37.1%) | | 50 (26.3%) | 11 (26.8%) | |
| IV | 10 (13.3%) | 14 (22.6%) | | 18 (9.5%) | 9 (22.0%) | |
| Missing | 0 (0.0%) | 3 (4.8%) | | 2 (1.1%) | 1 (2.4%) | |
| TNM Stage | | | 0.03 | | | 0.15 |
| I-II | 43 (57.3%) | 22 (37.3%) | | 120 (63.2%) | 20 (48.8%) | |
| III-IV | 32 (42.7%) | 37 (62.7%) | | 68 (35.8%) | 20 (48.8%) | |
| Missing | 0 (0.0%) | 3 (4.8%) | | 2 (1.1%) | 1 (2.4%) | |
| Death for CRC | | | 0.38 | | | 0.01 |
| No | 57 (76.0%) | 42 (67.7%) | | 167 (87.9%) | 29 (70.7%) | |
| Yes | 18 (24.0%) | 20 (32.3%) | | 23 (12.1%) | 12 (29.3%) | |

SE: standard error; FOBT: faecal occult blood test; gFOBT: guaiac faecal occult blood test; FIT: faecal immunochemical test; CRC: colorectal cancer; NA: not applicable.

Our results confirmed the positive impact of screening on CRC mortality risk specially with the implementation of the FIT. When compared with the screen-detected group, the risk of CRC death was 68% higher in the group with interval CRC and a two-fold risk of mortality was observed in the non-uptake group. Around 40% of all screen-detected cancers were diagnosed at stage I which adds support to the mortality findings. This difference in mortality rate was found when comparing the participants (screening vs interval) but not when the interval group was compared with the non-uptake group.

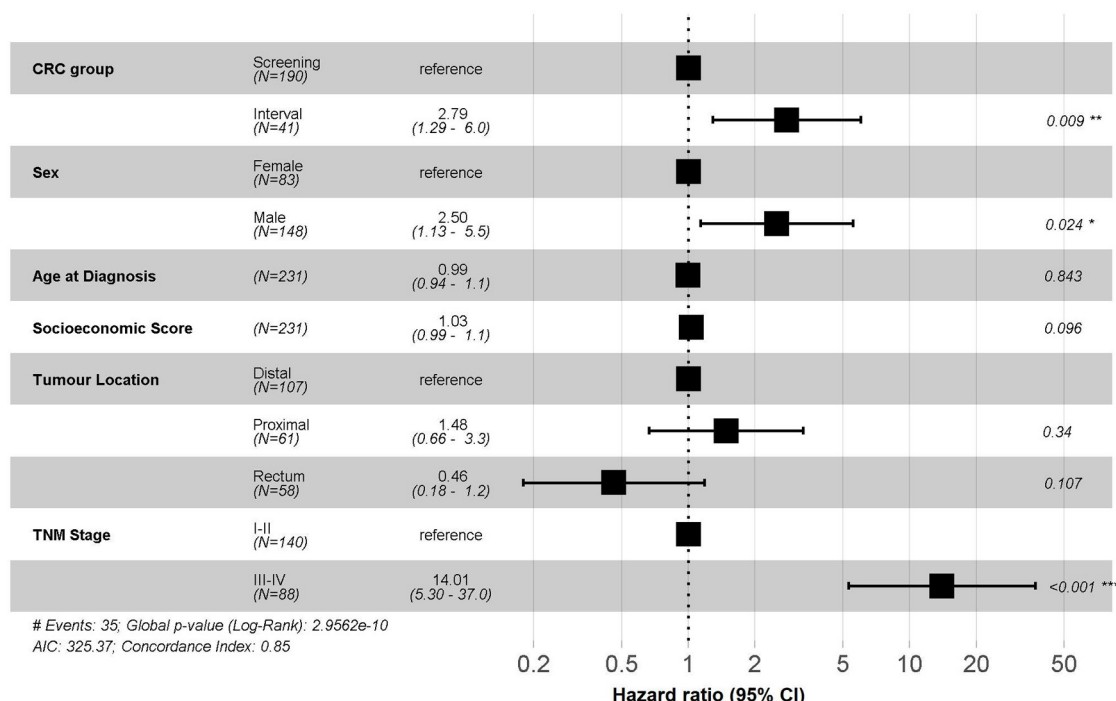

**Fig 3. Multivariate analyses for colorectal cancer mortality in FIT participants.** [a]Hazard ratios adjusted by the variables shown in this figure. CRC: colorectal cancer.

No randomized trial has been carried out on CRC screening using the FIT and the impact of the use of this test derives from observational studies with FIT and randomized trials with gFOBT [16–22]. However, there is sufficient evidence that the FIT meaningfully impact on CRC incidence and mortality [4, 11]. A recent systematic-review [23] has compared the mortality effect of CRC FOBT-screening (11 studies of gFOBT and 2 studies of FIT) across European regions. Among individuals invited to screening, CRC mortality was a 12% lower using the gFOBT and 36% lower when using FIT compared to those not invited. In our study, a higher reduction in mortality was achieved with the use of the FIT as a screening test as differences were found even when restricting for advanced-cancers and after considering lead-time bias. By contrast, the stratified analysis with gFOBT did not show an improved mortality rate among screen-detected CRCs.

There are two recent reviews that compare screen-detected versus FOBT interval cancer [9, 24]. First, Wieten et al. [24] published a systematic review and meta-analysis to evaluate the incidence rates of interval CRC following gFOBT and FIT in population-based CRC screening programmes. They reported a lower incidence rate of interval cancer after a negative FIT than after the gFOBT (20 vs 34 interval CRCs per 100,000 person-years, respectively). Afterwards, our group performed a review [9] focused only in FIT-based population screening and found that the interval CRC proportion was around 15% and that interval cancers were more frequent in women and in the right colon and were staged more advanced. In this study, the findings are consistent with the literature as the interval cancer proportion was 45.3% for gFOBT and 17.7% for FIT. And, when compared with the screen-detected group, interval CRCs were associated with rectum and proximal location and advanced TNM stage. Two Spanish investigations [25, 26] also reported an increased incidence of interval CRC located in the rectum. The FIT was better at detecting cancers in the distal colon; however, the mortality effect of the

FIT was similar regardless of the tumour location. In fact, a meta-analysis study reported that the FIT sensitivity for detecting CRC located in the distal colon was comparable to that in the proximal colon [27].

This reduction in mortality is in accordance with few studies that that have analysed the effect of the FIT in participants vs non-participants. Ventura et al. [28] observed a reduction in CRC mortality of 41% for participants as compared to non-uptakers. Idigoras et al. [26] reported a 5-year survival rate of 90.1% in screen-detected group, 76.3% in interval group and 60.5% in non-uptake group (P = <0.001). Corrado et al. [29] found an average overall survival in non-uptakers of 60.5%, whereas for participants, the overall survival was 85.1% (P = 0.001). Among the participating subjects, patients with interval cancers had worse overall survival rates (68.5%) than patients with screen-detected cancers (87.0%, P = 0.001).

Although we did not observe a higher frequency of CRC among men, male had a higher CRC mortality than women consistent with previous research [30]. However, our findings may be somewhat limited by the lack of information of patient comorbidities and received treatments. In our study population, men were older than women (p = 0.004) but we did not observe differences in the socioeconomic index, type of CRC detection, or tumour location. Another limitation of the study is that we were not able to assess the values of the faecal hae-moglobin of the FIT with a negative result prior to the diagnosis of interval cancer. By contrast, a strength of the study is the chart review performed to decrease misclassification of any participant or covariate analysed. Another strength is that we analysed a real-life scenario of a screening programme which undergoes a change of test. We have compared the impact on mortality of the use of the gFOBT and the FIT in the same screening population.

The current study data indicate that, with a participation rate of less than 50%, a reduction in mortality rates is achievable with FIT screening. This study shows how essential is achieving a high participation rate in the screening programme. Moreover, it is very important to ensure that people complete the screening process (having a conclusive FOBT result and completing colonoscopy after a positive FOBT) [31].

The group of interval cancers comprise a significant number of participants falsely reassured by their results. We should better inform our participants to carefully monitor the symptoms despite having a recent negative FOBT result.

In conclusion, our study confirms that there are differences in mortality according to the type of detection specially when using the FIT and these differences are not only explained by the delay in diagnosis.

## Supporting information

**S1 File.**
(DOCX)

## Acknowledgments

The members of the characterization and molecular subtyping of interval cancers in colorectal cancer screening research group (MSIC-SC) are as follows, in alphabetical order:

◦ Biobanc HUB-ICO-IDIBELL, L'Hospitalet de Llobregat, Barcelona, Spain: Susana López

◦ Biomarkers and Susceptibility Unit, Oncology Data Analytics Programme, Catalan Institute of Oncology (ICO), L'Hospitalet de Llobregat, Barcelona, Spain: Rebeca Sanz-Pamplona

◦ Cancer Screening Unit, Prevention and Cancer Control Programme, Catalan Institute of Oncology, L'Hospitalet de Llobregat, Barcelona, Spain: Gemma Binefa, Montse Garcia, Núria Milà, Carmen Vidal

◦ Catalonian Health Service (CatSalut), Barcelona, Spain: Elvira Torné

◦ Department of Gastroenterology, Bellvitge University Hospital, Hospitalet, Spain: Gemma Ibáñez-Sanz, Francisco Rodríguez-Moranta

◦ Department of Pathology, Bellvitge University Hospital (HUB-IDIBELL), L'Hospitalet de Llobregat, Barcelona, Spain: Xavier Sanjuan, Mar Varela

◦ Department of Pathology, Parc de Salut Mar, Barcelona, Spain: Mar Iglesias

◦ Department of Pathology, Hospital Clínic, Barcelona, Spain: Míriam Cuatrecasas

◦ Endoscopic Unit, Gastroenterology Department, Institut de Malalties Digestives i Metabòliques, Hospital Clínic, Barcelona, Spain: Maria Pellisé

◦ Oncology Data Science Group, Vall d'Hebron Institute of Oncology (VHIO), Barcelona, Spain: Fiorella Ruiz-Pace

◦ Prevention and Cancer Registry Unit, Service of Epidemiology and Evaluation, Parc de Salut Mar, Barcelona, Spain: Francesc Macià

◦ School of Nursing, University of Barcelona, Fundamental Care and Medical-Surgical Nursing Department, L'Hospitalet de Llobregat, Barcelona, Spain: Llúcia Benito

We thank CERCA Program, Generalitat de Catalunya for institutional support.

## Author Contributions

**Conceptualization:** Montse Garcia.

**Data curation:** Gemma Ibáñez-Sanz, Núria Milà, Judith Rocamora.

**Formal analysis:** Gemma Ibáñez-Sanz, Núria Milà.

**Funding acquisition:** Montse Garcia.

**Methodology:** Montse Garcia.

**Project administration:** Montse Garcia.

**Supervision:** Carmen Vidal, Víctor Moreno, Rebeca Sanz-Pamplona, Montse Garcia.

**Visualization:** Gemma Ibáñez-Sanz, Montse Garcia.

**Writing – original draft:** Gemma Ibáñez-Sanz, Montse Garcia.

**Writing – review & editing:** Gemma Ibáñez-Sanz, Núria Milà, Carmen Vidal, Judith Rocamora, Víctor Moreno, Rebeca Sanz-Pamplona, Montse Garcia.

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
