## [Decision Letter · Decision Letter 0]

4 Jun 2021

Positive impact of a faecal-based screening programme on colorectal cancer mortality risk

PONE-D-21-15357

Dear Dr. Garcia,

We’re pleased to inform you that your manuscript has been judged scientifically suitable for publication and will be formally accepted for publication once it meets all outstanding technical requirements.

Kind regards,

Ajay Goel

Academic Editor

PLOS ONE

Additional Editor Comments (optional):

Reviewers' comments:

Reviewer's Responses to Questions

**Comments to the Author**

1. Is the manuscript technically sound, and do the data support the conclusions?

Reviewer #1: Yes

2. Has the statistical analysis been performed appropriately and rigorously? 

Reviewer #1: Yes

3. Have the authors made all data underlying the findings in their manuscript fully available?

Reviewer #1: Yes

4. Is the manuscript presented in an intelligible fashion and written in standard English?

Reviewer #1: Yes

5. Review Comments to the Author

Reviewer #1: The work is well carried out and provides important information in the context of CRC screening programs and their impact on mortality.

Authors analyse survival data from a retrospective cohort of CRC screening program invitees (85000 people). They compare screen-detected CRC, interbal cancer and non-uptake cancer groups.

Authors show that CRC screening, mainly with FIT, is associated to a significant reduction in CRC mortality.

Statistical analysis are well performed and provided data support the conclusions.

6. PLOS authors have the option to publish the peer review history of their article (what does this mean?). If published, this will include your full peer review and any attached files.

Reviewer #1: No

---

## [Editor Report · Acceptance letter]

21 Jun 2021

PONE-D-21-15357 

Positive impact of a faecal-based screening programme on colorectal cancer mortality risk 

Dear Dr. Garcia:

I'm pleased to inform you that your manuscript has been deemed suitable for publication in PLOS ONE. Congratulations! Your manuscript is now with our production department. 

Kind regards, 

on behalf of

Dr. Ajay Goel 

Academic Editor

PLOS ONE